# Characteristic Parameters of Magnetostrictive Guided Wave Testing for Fatigue Damage of Steel Strands

**DOI:** 10.3390/ma16155215

**Published:** 2023-07-25

**Authors:** Xiaohui Chen, Jiang Xu, Yong Li, Shenghuai Wang

**Affiliations:** 1School of Mechanical Engineering, Hubei University of Automotive Technology, Shiyan 442002, China; 15286889608@163.com (X.C.); shwangkb@163.com (S.W.); 2School of Mechanical Science and Engineering, Huazhong University of Science and Technology, Wuhan 430074, China; m201770427@hust.edu.cn

**Keywords:** steel strand, fatigue life, nonlinear coefficient, attenuation coefficient, notch frequency, guided waves

## Abstract

Steel strands are widely used in structures such as bridge cables, and their integrity is critical to keeping these structures safe. A steel strand is under the working condition of an alternating load for a long time, and fatigue damage is unavoidable. It is necessary to find characteristic parameters for evaluating fatigue damage. In this study, nonlinear coefficients and attenuation coefficients were employed to evaluate fatigue damage based on magnetostrictive guided wave testing. Unlike pipe and steel wire structures, there is a phenomenon of a notch frequency when guided waves propagate in steel strands. The influence of the notch frequency on the nonlinear coefficient and attenuation coefficient is discussed. The relationship between the nonlinear coefficient, attenuation coefficient, and cyclic loading times was obtained through experiments. The amplitudes of the nonlinear coefficient and attenuation coefficient both increased with the increase in cyclic loading times. The experiments also showed the effectiveness of using these two characteristic parameters to evaluate fatigue damage.

## 1. Introduction

Steel strands, serving as the primary load-bearing constituents in prominent infrastructure systems, like cable-stayed bridges and prestressed concrete beams, endure prolonged cyclic loads, thereby resulting in the generation of internal fatigue damage [1,2]. Consequently, the damage gives rise to fractures and stress reduction in the strands, which affect the operational safety of the overall facilities. The conventional guided wave detection techniques employ flaw echoes to detect macroscopic damage in steel strands [3,4,5,6]. However, the fatigue damage in steel strands originates from microcracks that develop internally and propagate outward. Due to the inherent limitations of the wavelength of guided waves, the practical application of conventional guided wave technology in detecting microcracks in steel strands becomes arduous [7].

The occurrence of fatigue damage within a structural element induces inhomogeneous lattice or lattice defects within the material. When guided waves propagate through the fatigued element, they interact with these microscopic defects, resulting in waveform distortion, which causes energy transfer from low to high frequencies, and the excitation of high harmonics that generate nonlinear effects [8,9,10]. Notably, the second harmonic amplitude serves as a reliable indicator of member fatigue damage [11,12] and finds widespread application in characterizing early fatigue damage in metallic and composite structures. Rauter [13] employed the second harmonic generated therein to assess micro-damaged structures, monitoring damage progression using the nonlinear coefficient. The findings revealed that the nonlinear effect intensifies with the increasing size and number of microcracks, as evidenced by an augmented nonlinear coefficient. Similarly, Shen [14] observed that the nonlinear interaction between guided waves and fatigue cracks generates higher harmonic components, whereas linear interaction with notches fails to produce such nonlinear phenomena. Zhao [15] proposed an ultrasonic guided wave method that combines linear and nonlinear ultrasound, leveraging the phase velocity, elastic modulus, and nonlinear second harmonic to evaluate fatigue damage in composite materials. Furthermore, Meyer [16] employed guided wave technology to monitor fatigue damage in stainless steel pipes, demonstrating that guided waves exhibit heightened sensitivity to the presence of microcracks.

Ultrasonic guided wave technology has been utilized for detecting fatigue damage in aerospace industry components. Fromme [17] quantified the amplitude of the scattered wave by measuring the change in the incident wave’s angle with the defect direction and depth, thus detecting fatigue cracks in aerospace structures. Xu [18] conducted fatigue tests on aircraft structures and employed guided wave technology to diagnose fatigue cracks. The results demonstrated that guided wave technology can effectively mitigate the impact of uncertainty on fuzzy control. Masserey [19,20,21,22] conducted studies on a special high-frequency ultrasonic guided wave and found its high applicability in early fatigue damage detection and fatigue crack growth monitoring in aircraft components. Lee [23,24] achieved quantitative detection of fatigue cracks in steel joints under vibration using nonlinear ultrasound.

Previous research efforts focused on fatigue detection in steel wires, proposing the utilization of the coupling efficiency of magnetostrictive guided waves to discern fatigue damage in steel wires [25]. The findings revealed a reduction in coupling efficiency during the excitation and reception processes of magnetostrictive guided waves due to fatigue damage. Moreover, the group velocity of guided waves was observed to be jointly influenced by the tension and fatigue damage in the steel wire [26]. Additionally, the relationship between the group velocity and the stress and fatigue life of the wire was investigated and fitted. It is worth noting that steel strands possess a more intricate structure compared with steel wires. The radial contact between steel strand wires results in friction, leading to friction fatigue [27]. It also causes guided waves within the strand to exhibit notch frequencies [28,29,30,31]. Through a series of experiments, we discovered that after 1.85 million cycle loading times, the notch frequency exhibited an approximately linear correlation with the increase in the cyclic loading times [32].

The reliance on a single characteristic quantity to approximate the degree of fatigue damage in a steel strand is insufficient. Thus, this study aimed to explore additional characteristic parameters that can effectively characterize fatigue damage in steel strands. Recognizing that the fatigue damage of the strand influences its notch frequency, the impact of the notch frequency on the nonlinear coefficient was analyzed. To evaluate the fatigue damage using the nonlinear coefficient, it is crucial to avoid the notch frequency and its corresponding higher-order frequencies during guided wave excitation. Based on this foundation, an experimental investigation was conducted to further explore the relationship between the cycle loading times of the strand and the nonlinear coefficient. The experimental results indicated a positive correlation between the cycle loading times of the strand and the nonlinear coefficient. Moreover, considering the occurrence of fatigue damage, the presence of internal microcracks leads to increased scattering attenuation and absorption of waves during propagation, consequently altering the ultrasonic wave attenuation coefficient. The relationship between the cycle loading times of the steel strand and the attenuation coefficient of guided wave propagation within the strand was also examined. The results revealed a positive correlation between the cycle loading times of the steel strand and the wave attenuation coefficient. Both the nonlinear coefficient and the attenuation coefficient can serve as characteristic parameters for assessing fatigue damage in the strand.

## 2. Theoretical Background

A component with fatigue damage can result in unevenness in the internal lattice or the occurrence of lattice defects. In such cases, when ultrasonic waves propagate through the fatigue-damaged component, waves interact with micro-defects. This interaction causes waveform distortion, with the sound wave energy shifting from a low frequency to a high frequency, resulting in the excitation of higher harmonics and the generation of nonlinear effects. In contrast, when ultrasonic waves pass through a non-defective component, the waveform of the received signal remains undistorted, and the frequency of the received signal matches the frequency of the excitation signal. However, in the case of a component with fatigue damage, the received signal waveform is distorted and higher harmonics are generated. This disparity in the received signals between non-defective and fatigue-damaged components highlights the difference between traditional linear ultrasonic detection technology and nonlinear ultrasonic detection technology, wherein the frequency of the received signal deviates from the frequency of the transmitted signal.

The nonlinear coefficient, which characterizes the extent of nonlinear effects, can be expressed as the ratio between the amplitude of the fundamental wave and the amplitude of the second harmonic and is given by
(1)$$\beta =\frac{A_{2}}{A_{1}{}^{2}}$$
where $A_{1}$ represents the amplitude of the fundamental wave and $A_{2}$ is the amplitude of the second harmonic. The nonlinear coefficient is a characteristic parameter used for detecting fatigue damage in components. To calculate the nonlinear coefficient, a fast Fourier transform is performed on the received signal, which is obtained by measuring the amplitudes of the fundamental wave and the second harmonic. Based on the calculation of the nonlinear coefficient of the strand wires, it was observed that a notch frequency exists. The notch frequency does not correspond to the fundamental wave or the second harmonic frequency. This phenomenon can be explained as follows: If the notch frequency corresponds to the fundamental wave, the amplitude of the fundamental wave ($A_{1}$) will be small, resulting in a larger nonlinear coefficient. On the other hand, if the notch frequency corresponds to the second harmonic frequency, the amplitude of the second harmonic ($A_{2}$) will be too small, leading to a smaller nonlinear coefficient. Hence, the frequency corresponding to the fundamental wave or the second harmonic should be avoided as the notch frequency, as it can affect the accuracy of the measurement. Additionally, it is advisable to avoid the notch frequency coinciding with the frequency corresponding to the second harmonic when selecting the excitation parameters for testing the steel strand. This precaution helps in obtaining reliable and accurate results in the evaluation of fatigue damage.

The attenuation coefficient of ultrasonic waves in a component can be expressed as
(2)$$P_{x}=P_{0}e^{-\alpha x}$$
where *x* is the propagation distance, *α* is the attenuation coefficient, $P_{0}$ is the initial sound pressure amplitude, and $P_{x}$ is the sound pressure amplitude when the propagation distance is *x*. From Equation (2), it can be observed that the attenuation of ultrasonic sound pressure amplitude follows an exponential relationship with the propagation distance. It should be noted that the excitation frequency of the steel strand may affect the accuracy of the attenuation coefficient. To ensure accurate detection of fatigue damage in the steel strand using the attenuation coefficient, it is crucial to avoid using the excitation frequency that corresponds to the notch frequency.

## 3. Experiment Setup

A steel strand is formed by twisting together a center steel wire and six peripheral steel wires in a spiral cycle structure, resulting in mutual contact. When under tension, there is a certain amount of contact pressure between the steel wires. Due to the steel wire being generally drawn, its cross-section is not ideally round, exhibiting a certain degree of roughness on the surface. Consequently, in the axial tensile state, there will be increased friction between the steel wires, leading to friction fatigue. The typical cross-section of the strand after stressing is shown in Figure 1.

The material of the steel strands used was SWRS82B Φ 11 mm wire, the chemical composition of which is shown in Table 1.

A 1000 kN electro-hydraulic pulsating fatigue testing machine was employed to cyclically load the steel strands. The specimens consisted of three strands with seven epoxy-coated ASTMA416-270 grade low-relaxation wires each from the same batch, with a nominal diameter of 15.2 mm, each having a length of 4 m and a nominal tensile strength of 1860 MPa. The installation of the steel strands on the fatigue testing machine is shown in Figure 2, with the upper and lower ends clamped securely. Prior to installation, the solenoids of the transmitter and the receivers were mounted on the steel strands using single-layer 40-turn 0.5 mm enameled wire.

For distinction, the three steel strands were labeled as P01, P02, and P03, as shown in Figure 3, which depicts the fixture composed of clips and anchor plates. To ensure that the steel strands did not slip during stretching, the clamp blocks were designed with a tapered shape. The steel strands were first placed in the clamp blocks and then inserted into the sleeves. During the test, the clamps with the baffle were fixed onto the fatigue testing machine, as depicted in Figure 4. In order to prevent slippage of the steel strands during the loading process, preloading was applied using the fatigue testing machine. The tensile force was gradually increased to the upper limit stress and held for approximately half an hour. If no slip or abnormal behavior was observed, the tensile force was then unloaded to zero. If slippage occurred, the installation of the clamp was checked for correction. After the preloading process was completed, the cyclic loading started. Sinusoidal loading was employed as the loading mode with a frequency of 4 Hz. The ultimate stress applied on the strands during the experiment was set at 45% of the nominal tensile strength, corresponding to 837 MPa (117 kN), with a stress amplitude of 250 MPa (35 kN).

The installation position of the sensors on the steel strands is presented in Figure 5, with the clamping areas on both sides. The transmitter and the receivers were positioned 1000 mm apart from the left end, with distances of 600 mm, 850 mm, and 1100 mm between them, which were recorded as positions 1 to 3, respectively. Prior to conducting the cyclic loading experiment, a guided wave experiment was performed on the stranded wires, collecting data at positions 1, 2, and 3, respectively. The excitation signal used in the experiment was 5 cycles of an 80 kHz sinusoidal signal modulated by a Hanning window. Guided wave testing was conducted on the steel strands when the cyclic loading times reached 0.27 M, 0.58 M, 0.88M, 1.61 M, and 1.85 M, with a tensile stress of 78 kN applied during the detection. To accurately capture the change in the guided wave detection signal, the data were collected every 0.05 M times until a total of 39 sets of data were obtained at 3.94 M times. In order to eliminate the influence of the notch frequency on the nonlinear coefficient and the attenuation coefficient, the frequency of the excitation signal used in the experiment was adjusted according to the number of cyclic loading cycles.

## 4. Results and Discussion

### 4.1. Relationship between the Nonlinear Coefficient and the Cyclic Loading Times of the Steel Strands

The detection signals from three different loading stages of the unbroken P01 chain were taken as examples. The passing signal underwent a fast Fourier transform, and Figure 6 illustrates the frequency domain signals of the three cyclic loading stages. The second harmonic amplitude and fundamental amplitude of the three loading times were obtained and then substituted into Equation (1) to derive the nonlinear coefficients corresponding to different levels of fatigue. The relationship between the cyclic loading times and the nonlinear coefficients is depicted in Figure 7.

The relationship could be divided into two segments: 0–0.88 M times and 1.61 M–3.94 M times. The change in the nonlinear coefficient of the three steel strands from 0 times–0.88 M times was relatively smooth, with an overall increasing trend with the number of cyclic loading. Similarly, between 1.61 M times and 3.94 M times, the nonlinear coefficient of the P01 steel strand showed an increasing trend, with two segments from 1.61 M times to 3.03 M times, and from 3.13 M–3.94 M times. The change in the P02 steel strand between 1.61 M times and 3.00 M times was relatively smooth, followed by an increasing trend in the relative nonlinearity after 3.00 M times, with a faster rate of change. The nonlinear coefficient of the P03 steel strand also increased with the number of cyclic loading times.

To quantitatively assess the fatigue damage of the strand, the relationships between the cyclic loading times and the nonlinear coefficients of the P01 steel strands were normalized and then subjected to exponential fitting. The normalization process was also applied to the P02 and P03 steel strands. The normalized cyclic loading time and nonlinear coefficients fitting curve is presented in Figure 8, showing that the trend of the detected data distribution for the P02 and P03 strands was approximately consistent with the fitted curves. The equation for the normalized cyclic loading time and nonlinear coefficient fitting is represented as Equation (3).
(3)$$y_{N}=0.0001565e^{4.452x}+0.1188e^{0.4928x}$$
where $y_{N}$ is the normalized nonlinear coefficient and the unit of x is ten thousand times.

The results indicated that guided waves propagated through the fatigue-damaged steel strands and interacted with microcracks within the strands, resulting in a nonlinear effect and an increase in the nonlinear coefficient. Therefore, the nonlinear coefficient can be used as a characteristic parameter for magnetostrictive guided wave detection of fatigue damage in steel wires, once the excitation parameter avoids the notch frequency that corresponds to the second harmonic. 

### 4.2. Relationship between the Cyclic Loading Times and the Attenuation Coefficients

Taking the detection signals of three positions of the unfatigued P01 strand as an example, Figure 9 illustrates the time-domain signals of three positions. When calculating the attenuation coefficient, Equation (2) was combined with the pass signals at three positions, and the peak-to-peak value of the pass signals at these positions was fitted using an exponential function. Figure 10 presents the attenuation coefficient of the steel strand under different cyclic loading times. Figure 11 displays the normalized cyclic loading times and attenuation coefficient curves for P01, P02, and P03, as well as the fitting curve for the P01 strand. Equation (4) presents the fitting equation for the normalized cyclic loading times and attenuation coefficients. The relationship could also be divided into two stages: 0–1.61 M times and 1.85 M–3.94 M times. Between zero times and 1.61 M times, the P01 steel strands exhibited an increasing trend. While the P02 steel strand showed an increasing trend from 0–0.88 M times, the change was relatively flat at 1.61 M times, with a large value at this point. Between 1.85 M times and 3.94 M times, the attenuation coefficients of the P01 and P02 steel strands increased with the cyclic loading times, and the attenuation coefficients of the two steel strands were relatively close. However, the attenuation coefficient fluctuations of the P03 steel strand were larger than those of the P01 and P02 steel strands, reaching a maximum value at 2.67 M times, and then decreasing to a minimum value at 3.82 M after 3.73 M times. It may be that as the cyclic loading times increased, the notch frequency of the guided wave in the strand changed. From the overall trend, the attenuation coefficient of the three steel strands increased with the number of cyclic loading times.
(4)$$y_{A}=0.127e^{0.4906x}+0.005933e^{-0.6199x}$$
where the $y_{A}$ is the normalized attenuation coefficient and the unit of x is ten thousand times. The results in Figure 11 indicate that the overall trend of the wave attenuation coefficient in the three steel strands increased with the number of cyclic loading times. This finding suggested that fatigue damage in the steel strand due to the generation of microcracks led to an increase in wave attenuation during propagation, resulting in an increase in the attenuation coefficient. Therefore, the attenuation coefficient can serve as a characteristic parameter for the detection of magnetostrictive guided waves in steel wire fatigue damage after avoiding the notch frequency in the excitation parameter. Furthermore, although the overall trend of the three curves in Figure 11 is increasing, some exhibit significant fluctuations, such as the attenuation coefficient after 3.73 M cycles of loading of the P03 steel strand. For the occurrence of abnormal results, the influence of the notch frequency should be considered again to correct the results.

Figure 12a shows the cross-sections of an unfatigued strand with no contact between peripheral wires. And Figure 12b shows the cross-sections of the same strand after 3.94 M cycles of loading, with contact between peripheral steel wires and deformation at the contact area.

We continued to apply cyclic loads to the strand until one of the wires fractured, and its fracture cross-section is shown in Figure 13.

Combining the guided wave detection results with Figure 13, it is evident that when the strand was subjected to a certain level of cyclic load, small cracks were generated inside its steel wire. If the cyclic load continued to be applied, these small cracks were enlarged until a material fracture occurred. 

## 5. Conclusions and Future Work

The nonlinear coefficient and the attenuation coefficient obtained from the magnetostrictive guided wave testing signal were employed to evaluate the fatigue damage of steel strands. This research is relevant for detecting fatigue damage in seven-core strands subjected to axial cyclic loading under a fixed tensile force. It is essential to ensure that there are no visible corrosion or fracture defects on the strands. The results showed the following:(1)The nonlinear coefficients varied with the number of cyclic loading times. In the range of 0 to 3.94 M cycle loading times, the relationship between the number of cyclic loading times and the nonlinear coefficients showed an approximate exponential growth.(2)The attenuation coefficient varied with the number of cyclic loading times. In the range of 0 to 3.94 million cycle loading times, the relationship between the number of cyclic loading times and the attenuation coefficient showed an approximate exponential growth. However, when the number of cyclic loading times reached 3.73 million, the attenuation coefficient exhibited a decreasing tendency due to the influence of the notch frequency of steel strands.(3)When using the nonlinear coefficient and the attenuation coefficient to evaluate the fatigue damage in steel strands, the excitation parameters should avoid the notch frequency and its higher harmonic.

Due to limitations in conducting cyclic loading experiments, we have performed experiments only under fixed tension on the strand so far. Initially, we obtained the relationship between the nonlinear coefficient of guided waves and the attenuation coefficient, and the fatigue life of the strand in the range of 0–3.94 M cycles, which verified the potential of the nonlinear coefficient and the attenuation coefficient as characteristic parameters for predicting the fatigue life of the steel strand. In the future, we plan to conduct cyclic loading experiments under various tensions. Additionally, we will investigate the relationship between the nonlinear coefficient and attenuation coefficient with the cyclic loading times of the strand under different tensile forces.

In addition, since the notch frequency may change with the fatigue state of the steel strand, it is necessary to use the missing frequency to make necessary corrections to the detection results. Subsequent research will be carried out on the correction method in order to establish a more accurate relationship to support the assessment of fatigue damage.

## Figures and Tables

**Figure 1 materials-16-05215-f001:**
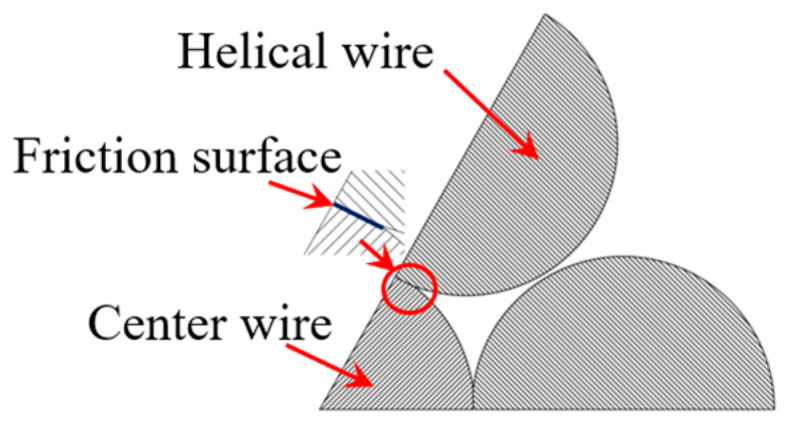
Typical strand cross-section after stressing.

**Figure 2 materials-16-05215-f002:**
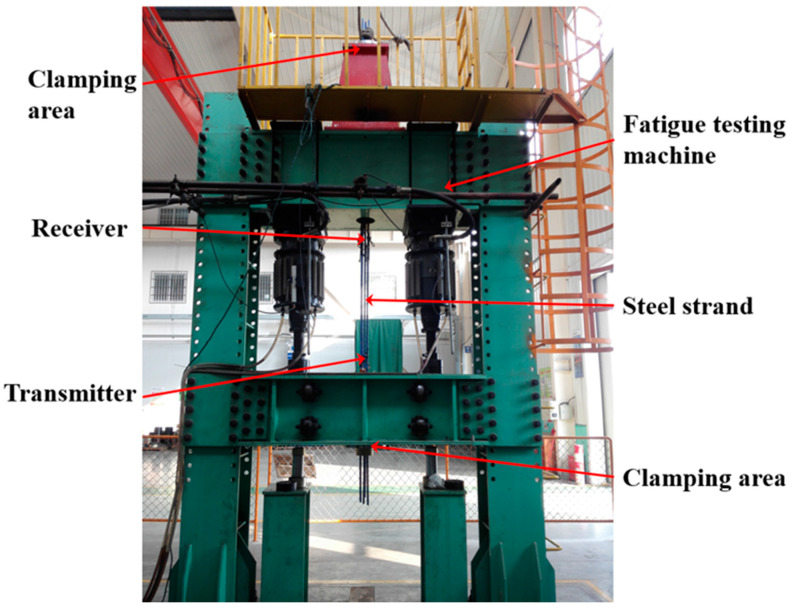
Photo of the steel wire loading experiment installation.

**Figure 3 materials-16-05215-f003:**
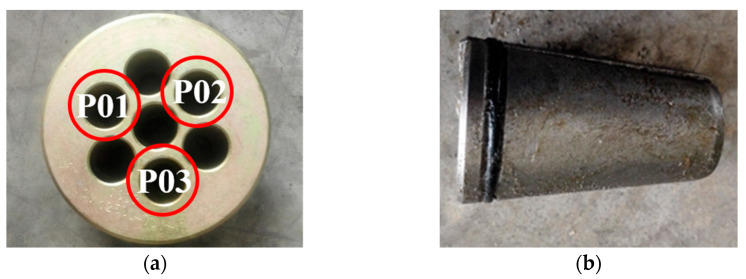
Photo of the steel strand fixture and the numbers of steel strands: (**a**) anchor plate; (**b**) clips.

**Figure 4 materials-16-05215-f004:**
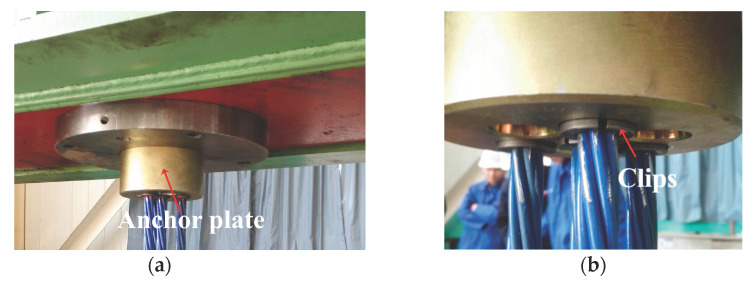
Photo of the installation of the steel strand fixture: (**a**) anchor plate; (**b**) clips.

**Figure 5 materials-16-05215-f005:**
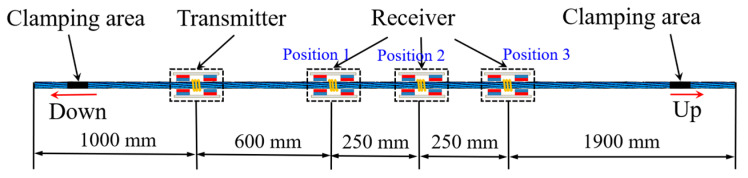
Schematic diagram of the installation position of the sensor in the cyclic loading experiment.

**Figure 6 materials-16-05215-f006:**
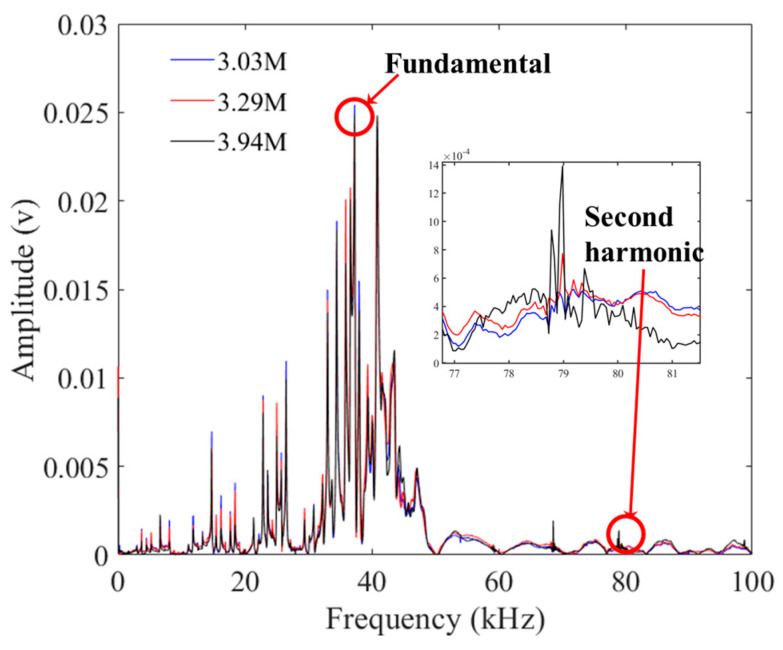
The frequency domain signals of three cyclic loading stages.

**Figure 7 materials-16-05215-f007:**
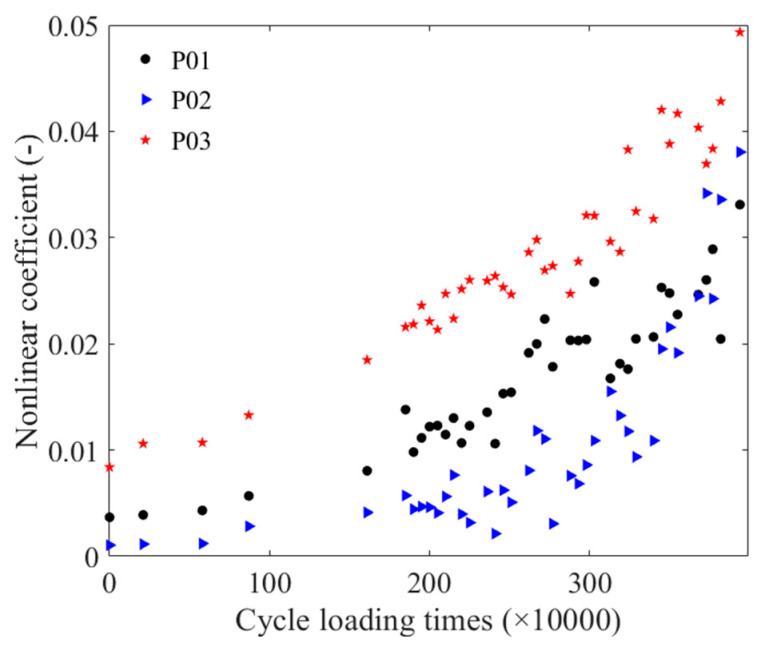
The relation between the cyclic loading times and the nonlinear coefficients.

**Figure 8 materials-16-05215-f008:**
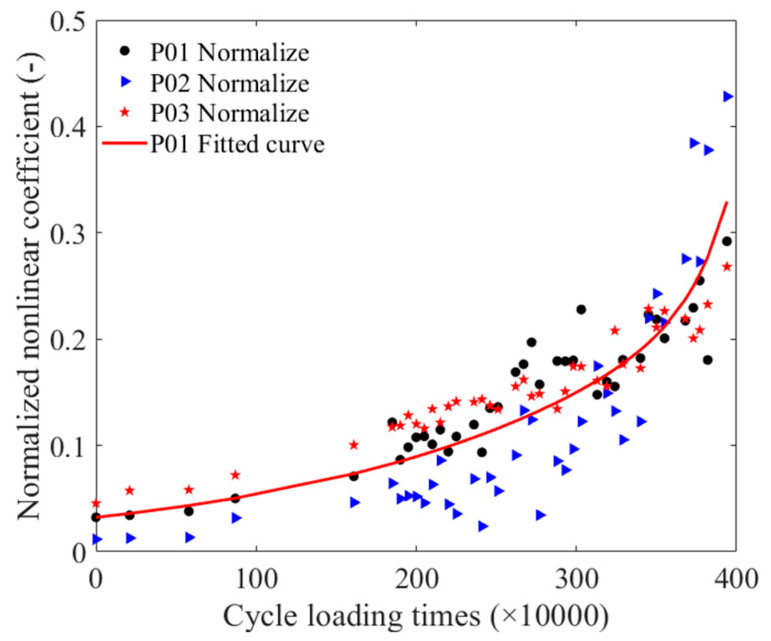
The normalized cyclic loading time and nonlinear coefficients fitting curve.

**Figure 9 materials-16-05215-f009:**
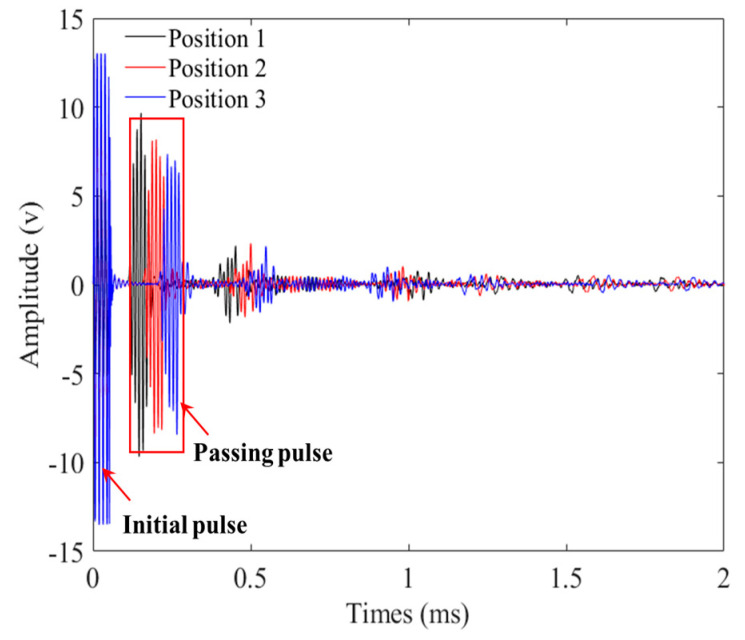
Time-domain waveform obtained from three positions of an unfatigued steel strand.

**Figure 10 materials-16-05215-f010:**
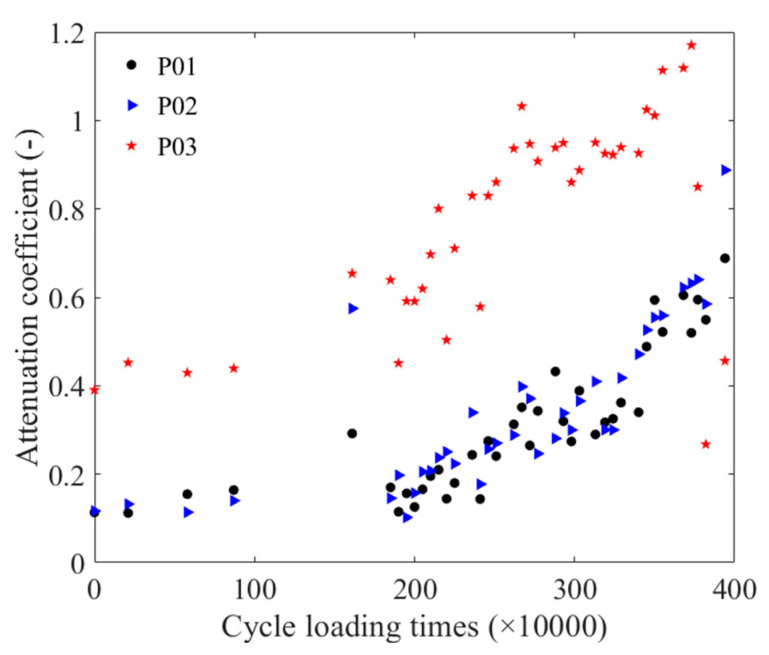
Relationship between the number of cyclic loading times and the attenuation coefficient.

**Figure 11 materials-16-05215-f011:**
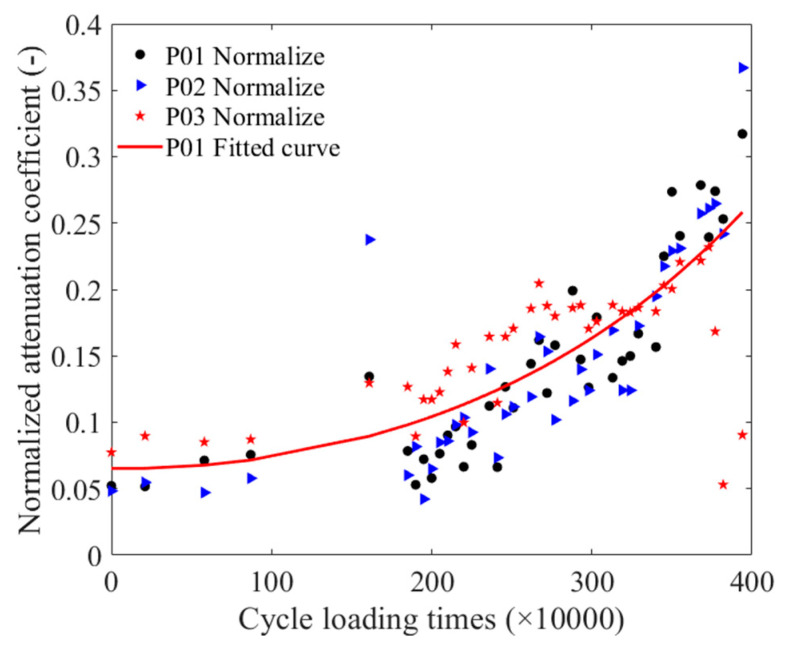
The normalized cyclic loading time and attenuation coefficient fitting curve.

**Figure 12 materials-16-05215-f012:**
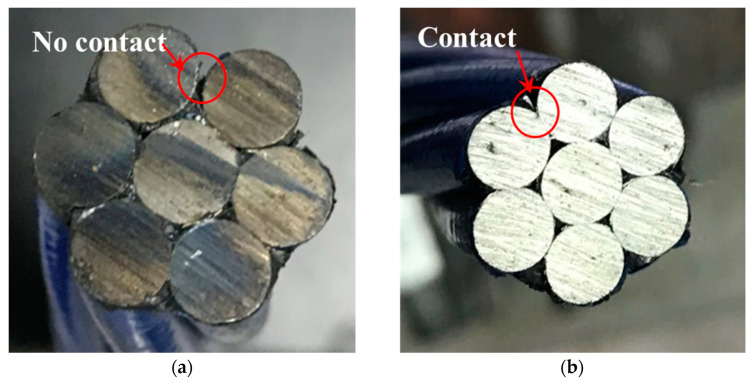
The cross-sections: (**a**) unfatigued; (**b**) fatigued after 3.94 M cycles.

**Figure 13 materials-16-05215-f013:**
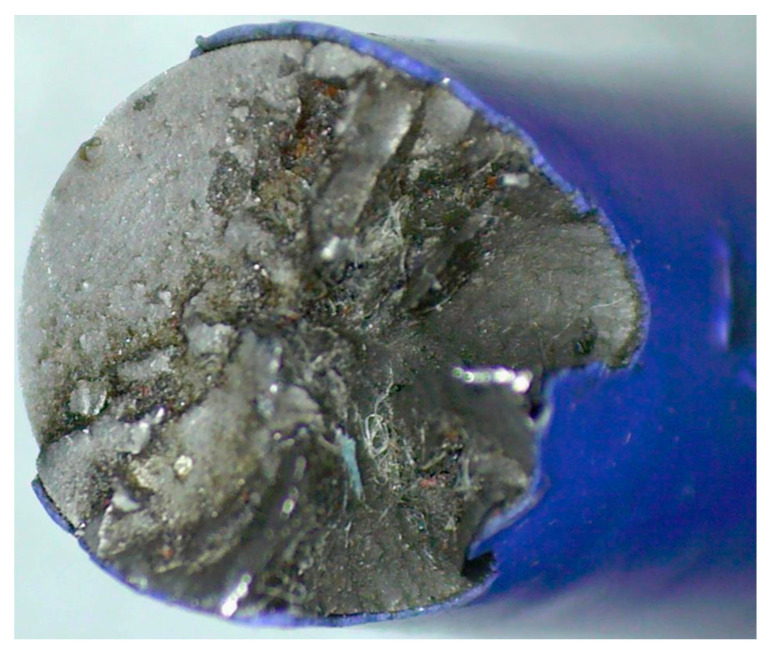
Photo of a fatigued wire fracture.

**Table 1 materials-16-05215-t001:** The chemical composition of SWRS82B wire (wt%).

	C	Si	Mn	S	P	Cr	Ni	Cu
Content (wt%)	0.83	0.23	0.76	0.007	0.01	0.2	0.03	0.09

## Data Availability

The data supporting the reported results can be sent by the authors via e-mail.

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
