# Peer review of "Characteristic Parameters of Magnetostrictive Guided Wave Testing for Fatigue Damage of Steel Strands"

_materials, 2023, doi:10.3390/ma16155215_

Round 1

Reviewer 1 Report

The paper focus on analyzing the relationship between the cycle loading times and attenuation coefficient in the damage behavior of steel strands.

Experiments of three samples were performed in a steel wire loading experiment installation with sensors located along the test sample to capture the change in the guided wave.  

Results showed a relationship between nonlinear coefficients and the cycle loading times, which increase as the cycle loading time increase. Those results are in agreement with other authors.

The topic is relevant and the reviewer considers it to be an adequate contribution that aligns well with the scope of journal materials. However, improvements are necessary since the paper is too short and lacks discussion.

I recommend showing the fatigue register, such as load time and/or the number of cycle-failure. Relate the signal with material properties, maybe damage density or remaining life, and include it in the discussion. 

Line 14 – remove “the” before magnetostrictive

Line 18 – the acrylic

Line 30 – conventional two times

Lin 147 – The tensile force

Line 14 – remove “the” before magnetostrictive

Line 18 – the acrylic

Line 30 – conventional two times

Lin 147 – The tensile force

Author Response

Thank you for your comments. We have carefully considered your comments and have made revisions to address them, which are attached.

Reviewer 2 Report

Please refer to the comments/remarks listed below:

Author Response

(The authors gave the same response as above.)

Reviewer 3 Report

(1) The paper assessed the fatigue response of steel strands by means of guided wave testing. In particular, experimental tests are carried out on three strand specimens, applying a sinusoidal cyclic protocol. The results are assessed in terms of (a) nonlinear coefficient (ratio between the second harmonic amplitude and the squared fundamental wave amplitude) and (b) attenuation coefficient (considering an exponential correlation).

(2) The paper is promisingly contributing to the field literature, but the referee only has a concern, discussed in the following. The results are reported only in terms of direct features, correlated with the cycles. These results may be interesting, but they do seem not fully sufficient for an original scientific publication. More advanced elaborations should be carried out (e.g., statistical-based analysis) and damage assessment/detection criteria should be possibly developed, even though following a preliminary approach. The detected parameters should be clearly correlated to the fatigue damage/response. Furthermore, minor revisions should also be implemented, as reported below.

·       Check line 30 (“Conventional the conventional”).

·       Please, avoid the use of “damages” (line 31) since this word typically refers to a meaning different from “damage”.

·       The literature review and the cited references are relatively poor despite several studies assessing the fatigue response of metallic structures and components through non-destructive techniques. Please, extend the literature review/references. Some potentially relevant studies are reported in the following.

o   Cho, S.-M., Thomas, B.G., Hwang, J.-Y., Bang, J.-G., Bae, I.-S., 2021. Modeling of Inclusion Capture in a Steel Slab Caster with Vertical Section and Bending. Metals 11, 654. https://doi.org/10.3390/met11040654

o   D’Angela, D., Ercolino, M., Bellini, C., Di Cocco, V., Iacoviello, F., 2021. Failure criteria for real-time assessment of ductile cast irons subjected to various loading conditions. Smart Mater. Struct. 30, 017001. https://doi.org/10.1088/1361-665X/abc56f

o   Debeleac, C., Nastac, S., Musca (Anghelache), G.D., 2020. Experimental Investigations Regarding the Structural Damage Monitoring of Strands Wire Rope within Mechanical Systems. Materials 13, 3439. https://doi.org/10.3390/ma13153439

o   Ji, Q., Jian-Bin, L., Fan-Rui, L., Jian-Ting, Z., Xu, W., 2022. Stress evaluation in seven-wire strands based on singular value feature of ultrasonic guided waves. Structural Health Monitoring 21, 518–533. https://doi.org/10.1177/14759217211005399

o   Kurz, J.H., Laguerre, L., Niese, F., Gaillet, L., Szielasko, K., Tschuncky, R., Treyssede, F., 2013. NDT for need based maintenance of bridge cables, ropes and pre-stressed elements. J Civil Struct Health Monit 3, 285–295. https://doi.org/10.1007/s13349-013-0052-5

o   Wang, Y., Li, Z., Zhu, X., Gong, Y., Liu, N., Deng, Q., Long, Z., Teng, J., 2023. Plane stress measurement on the cross-section of steel components using ultrasonic shear waves. Mechanical Systems and Signal Processing 191, 110185. https://doi.org/10.1016/j.ymssp.2023.110185

·       Please, report relevant reference associated with both nonlinear coefficient and attenuation coefficient formulations.

Minor editing of English language required.

Author Response

(The authors gave the same response as above.)

Reviewer 4 Report

The topic of the manuscript is very interesting. Unfortunately the authors didn;t succeed to address the proposed topic.

Sugestions from my side:

- the introduction part must be extended - present similar studies with the same topic, including the research results;

- materials and methods part is not detailed enough - please detail more the testing procedure, describing in depth the stardeds used for testing;

- the results part (the most important part) is almost inexistent. Please detail more this part additng comparisions between your study and other research findings; 

- the conclusion part must be rewritten.

In this form the manuscript cannot be published. It must be reconsidered and extended.

Minor spelling/typing errors. This part must be reassessed following the extension of the manuscript.

Author Response

(The authors gave the same response as above.)

Reviewer 5 Report

The article needs significant revision and addition of my comments:

Lines:

30. Conventional 2x, correct in the text

121. x - which x is it in the formula?

150. rewrite the word in the correct form - correction

159. units after the value must be next to each other (1000 mm) not on the second line, correct this in the entire text, the same for example line 167. kN or line 181. M

Fig.5 insert the legend on the edge Fig. 5 (in the corner), Enlarge the font in Fig. 5

Fig. 4 both ends of the rope have the inscription clamping area, it is not clear which end is up and down, or change the image from horizontal to vertical

230. change million instead of M

Fig. 2, the marking of the anchor plate does not match the marking on Fig. 3

160 and fig 4, according to what did they choose the distances?

205 – 207 Change times to mark M (184 – 188 times)

1. Increase the number of bibliographic references presented in a minimum amount of 30 considering the scope of the presented publication.

2. The list of bibliographic references does not contain the necessary details, especially the DOI number, so the form of entry does not correspond to the submitted template.

3. The conclusion section should clearly present the achieved results, we recommend presenting the results of the publication in the form of bullet points with clearly defined benefits of the work

4. Figure 2 and figure 3 should be supplemented with a) and b) and described in the text

5. For example, in line 222 of Fig.7, in the other lines, the images of Figure are marked. 7. unify the registration method

6. Figure 5 and 7 add units to the Y axis unless there is a unit to add (-).

7. Figure 1 white background under the description covers a significant area of the image (remove the colored background)

8. Since this is a journal focused on materials engineering, it is necessary to add a material chapter describing the chemical and mechanical properties of the investigated steel rope

9. Add evidence of the presence of microcracks using microscopy to the results

To check the quality of the English language, the large number of repetitions of some words, as well as the order of words and the inflection of words is insufficient.

Author Response

(The authors gave the same response as above.)

Round 2

Reviewer 2 Report

Dear Authors, I received a satisfactory answer to all my doubts. All my concerns about verifying the presented method based on measurement results, the presentation of your results and the conclusions were clarified. I have no critical remarks about the whole article; on this basis, I recommend its publication in its current form.

Author Response

Thank you!

Reviewer 3 Report

The revisions are noted, as well as the strengthening of the scientific aspects is acknowledged. The following minor revisions are required prior to considering the paper suitable for publication.

1. Please, provide a quantitative measure of the fitting quality for the developed correlations.

2. Please, provide quantitative criteria for expeditious assessment purposes, including applicability conditions (e.g., case study features).

3. Please, provide conclusive comments on the use and extension of the provided correlations in similar case studies and applications, stressing the limitations of the study (e.g., representativeness of the case study, number of tests)

English is fine. Please, implement a rapid grammar and syntax check prior to sending the final version of the paper.

Author Response

Thank you for your comments again!

Reviewer 4 Report

The authors revised the manuscript according with the reviewers comments.

Minor editing of English language required

Author Response

Thank you! 

We have checked the grammar of the article and corrected the errors carefully.

Reviewer 5 Report

Thanks to the authors for including my comments.

I still have the following small suggestions for editing the presented article:

1. Add to paragraph 3, a table with the chemical composition of the material from which the rope is made.

2. Pictures 7, 8, 10, 11 - on the X axis, change the text (the thousand times) to mathematical numerical notation.

3. Pictures 7, 8, 10, 11 - put a dash in the parentheses on the Y axis, even if these coefficients are unitless.

4. Figures 6-11 unify the size, color and style of the font on the axes.

5. Sentence on line 288-290,, As shown in Figure 12 (b), the existence of obvious friction marks on the steel wire surface also further verifies that the strand fatigue is mainly caused by friction fatigue,, How did you come to this claim? For this statement, you need an in-depth analysis of the occurrence of microcracks on the surface of the friction mark. And you don't have that anywhere.

I recommend taking another picture of picture 12b and focusing on the quality of the photo. The image should be taken at a higher magnification or keep image 12b but take another photo of the selected friction area at a much higher magnification.

6. It is also necessary to evaluate the type of wear that occurs on the surface friction mark.

7. Conclusion - Although the conclusions are written in three points, there are no specific results, the present ones describe only theoretical conclusions. It is necessary to enter exact values in the conclusions. It is also necessary, since this magazine is dedicated to material research, to insert the result from Fig. 12.

It is also necessary to write what type of fatigue fracture occurred? Was it a crystalline fracture or intercrystalline?

In this form, the article is written only at a slightly professional theoretical level. It is also necessary to confront your results with other authors. It would be appropriate if you inserted a short paragraph "discussion" in the article where you would compare your findings.

PS: I'm not an expert on the stylization of English grammar, but some phrases are wrong from my layman's point of view. I recommend having the article checked for grammar.

Author Response

Thank you for your comments again!
